

# Characteristics of Agricultural Droughts in CMIP6 Historical Simulations and Future Projections

Lukas Lindenlaub[1, 2], Katja Weigel[1, 2], Birgit Hassler[2], Colin Jones[3], and Veronika Eyring[2, 1]

[1]University of Bremen, Institute of Environmental Physics (IUP), Bremen, Germany
[2]Deutsches Zentrum für Luft- und Raumfahrt e.V. (DLR), Institut für Physik der Atmosphäre, Oberpfaffenhofen, Germany
[3]National Centre for Atmospheric Science and School of Earth and Environment, University of Leeds, Leeds, LS2 9TJ, United Kingdom

**Correspondence:** Lukas Lindenlaub (lindenlaub@uni-bremen.de)

**Abstract.** This study explores changes in agricultural drought event characteristics in projections of Earth System Models (ESMs) participating in the Coupled Model Intercomparison Project Phase 6 (CMIP6) for different future scenarios based on three Shared Socioeconomic Pathways (SSP). To quantify the intensity of agricultural droughts, the 6-month Standardized Precipitation Evapotranspiration Index (SPEI6) with a 65 year reference period is applied to the simulations of 18 ESMs.

In a first step, these ESMs are evaluated based on performance metrics and pattern correlations of drought related variables including precipitation and approximated reference evapotranspiration with reanalysis datasets including ERA5 and CRU. With this we extend the model benchmarking performed in the third chapter of the IPCC AR6 by 15 years and additional variables. In a second step we analyze global and regional projected SPEI6 distributions to estimate and characterize the changes in agricultural drought in the future based on multi-model means of change rates, distributions and relative area covered by

specific events. We quantify the change of drought index values for 42 IPCC AR6 WG1 reference regions individually with a focus on those with most harvest area and find negative trends in water budget and SPEI for higher emission scenarios in most of them, particularly in the Mediterranean and other arid regions. This agrees with other recent studies. Increasing reference evapotranspiration emerges as the dominant driver for drier conditions in these regions. What is considered as the driest 2.3 % months during 1950-2014 is projected to be the new normal or moderate condition in arid regions by 2100, following a high

emission future scenario (SSP5-8.5). For this scenario, 40 % of the harvest regions surface is considered to be under extreme drought conditions during northern hemisphere autumn. Under a low emission scenario (SSP1-2.6) with an expected global warming of 1.8 °C it would be less than 10 %. Our results show a significant difference between future scenarios regarding distribution shifts and spatial extend of extreme drought conditions in harvesting regions and can serves as a foundation for further impact and mitigation studies.

## 1   Introduction

The availability of water is an essential requirement for human life, and crucial for food production, freshwater supply and industry. If water is sparse, droughts occur which are dangerous and complex natural hazards. Droughts can affect more people than most other natural disasters (Hagman et al., 1984; Wilhite, 2000). Changes in their characteristics and their impacts in



a changing climate are still not fully understood. Considering a drought as an exceptional dry event requires some normal
climatic conditions as reference, which is one of the challenges in defining droughts. A definition further depends on the
field of application and can be roughly clustered into different types of drought. A decline of precipitation over several days
to months is often referred to as a meteorological drought. Low soil moisture and high plant water stress are indicators for
agricultural droughts. Ongoing precipitation deficit in combination with runoff and evapotranspiration can impact freshwater
reservoirs like lakes, rivers and groundwater, which are indicators for hydrological droughts (Wilhite, 2000).

Recent events like northern hemisphere mega drought in 2022 (Montanari et al., 2023; Xu et al., 2023; Schumacher et al.,
2022) or European multi year droughts since 2016 (García-Herrera et al., 2019; Rakovec et al., 2022) have been analyzed and
linked to changes in climate (Yu et al., 2023). The Sixth Assessment Report (AR6) of the Intergovernmental Panel on Climate
Change (IPCC) found an agreement in recent scientific research with at least medium confidence in an increase of agricultural
droughts for several regions of the world (Masson-Delmotte et al., 2021; Seneviratne et al., 2021). The recent increase is more
profound than detected for meteorological droughts and may implicate serious consequences for humans.

This motivates the investigation of the impact of changing climate conditions on the processes of formation and characteris-
tics of agricultural droughts in the future. In the Coupled Model Intercomparison Project Phase 6 (CMIP6; Eyring et al., 2016a)
future scenarios for green-house gas (GHG) emissions and land-use change based on Shared Socioeconomic Pathways (SSP)s
are described to be used in model projections. There is ongoing research in analyzing droughts using different methods and
metrics based on such pathways (Bakke et al., 2023; Vicente-Serrano et al., 2022; Balting et al., 2021; Zeng et al., 2021; Zhao
and Dai, 2022; Xu et al., 2019).

Studies focusing on (semi) global characteristics agree on a drying trend for the Mediterranean, Europe and parts of America
and southern Africa in higher emission scenarios according to the SPEI (Bakke et al., 2023; Vicente-Serrano et al., 2022; Zeng
et al., 2021) and other drought indices (Zhao and Dai, 2022; Tabari and Willems, 2022).

Here we contribute to this research effort by including the low emission scenario following SSP1-2.6 (expected global
warming of 1.8 K at the end of the century) in addition to SSP2-4.5 and SSP5-8.5 to an analysis of agricultural droughts
categorized by SPEI. We also use drought characteristics that are suited for a changing climate and discuss the observed
changes for IPCC AR6 reference regions with a focus on regions with most harvest area. Even though there are already
dedicated evaluations of the used ESM data, we provide a comprehensive evaluation of the 18 model ensemble with different
reanalysis data for the exact variables used in our analysis. The methods we use to achieve that follow recommendations
from the World Meteorological Organization (WMO), the Food and Agriculture Organization of the United Nations (FAO)
and American Society of Civil Engineers (ASCE). They are described in detail in Section 2. In Section 3 the selection of
models from the Coupled Model Intercomparison Project Phase 6 (CMIP6; Eyring et al., 2016a) and the datasets used for their
evaluation are listed and described. Section 4 describes the results splits into three parts. The first focus is on the evaluation of
predictions for the historical period 1950-2014 followed by the second part focusing on the temporal and spatial analysis of
SPEI in different regions and future scenarios. In the third part we discuss the limitations of the shown results.



## 2 Methods

### 2.1 Reference Evapotranspiration

The Reference evapotranspiration ($ET_0$) also referred to as Potential Evapotranspiration (PET) or Atmospheric Evaporative
Demand (AED) is a measure for the amount of water that the atmosphere can take up from a saturated reference surface with
unlimited water supply. Following Allen et al. (1998) we use watered crop land with an albedo of 0.23 as reference surface.

Over the last century a variety of methods have been developed to approximate the PET, including the methods introduced
by Thornthwaite (1948) and Hamon (1963). Although used often, both approximations are not reliable under changing climate
conditions (Shaw and Riha, 2011). In contrast, Equation 1 provides a more complex combination of parameters to approximate
$ET_0$ with a closer link to the underlying physical processes. This approximation is more adequate and recommended by FAO
and ASCE (Pereira et al., 2015; Allen et al., 1998; Technical Committee on Standardization of Reference Evapotranspiration,
2005). It is derived from Penman (1948) and Monteith (1965) and includes several variables like the soil heat flux density $G$,
mean daily air temperature at 2 m $T_d$, wind speed at 2 m $u_2$, the psychometric constant $\gamma$, saturation and actual vapor pressure
$e_s$ and $e_a$, such as net radiation at plants surface $R_n$ and heat flux density $G$. $\Delta$ is the slope of the vapor pressure curve (Allen
et al., 1998). Instead of 0.12 m short grassland we assume a crop height of 0.5 m following Walter et al. (2001). With adjusted
crop coefficients (1600 and 0.38) for monthly time steps this leads to the ASCE equation for standardized reference crop
evapotranspiration for tall surfaces $ET_{RS}$ (Technical Committee on Standardization of Reference Evapotranspiration, 2005):

$$ET_0 = ET_{RS} = \frac{0.408\Delta(R_n - G) + \gamma\frac{1600}{T+273}u_2(e_s - e_a)}{\Delta + \gamma(1 + 0.38u_2)} \tag{1}$$

Allen et al. (1998) and Zotarelli et al. (2010) provide detailed instructions on the calculation of the psychometric constant
and how to approximate variables that may not be available in observational or model data (i.e. $u_2$ from wind speed at 10 m,
$R_n$ from downwards short wave radiation and $T_d$ from monthly mean daily minimum and maximum temperature).

### 2.2 Standardized Precipitation Evapotranspiration Index

In contrast to its predecessor SPI, which is solely based on precipitation, the SPEI includes additional atmospheric variables
through potential or reference evapotranspiration $ET_0$. While the SPEI, as originally proposed by Vicente-Serrano et al. (2010),
uses only temperature data to approximate $ET_0$ (Thornthwaite, 1948), we make use of a more complex approximation described
above. The SPEI is calculated from $ET_0$ and $PR$ by fitting a probability distributions of water budget ($ET_0$-$PR$) over a chosen
reference period and transfer it to a standard distribution. In contrast to precipitation in the SPI the water budget can be negative,
therefore a three parameter distribution like the generalized log-logistic is used (Beguería et al., 2014):

$$F(D) = \left[1 + \left(\frac{\alpha}{D - \gamma}\right)^{\beta}\right]^{-1} \tag{2}$$

In Equation 2 $F$ is the probability distribution of a variable $D$ with $\alpha$, $\beta$ and $\gamma$ being scale, shape and location parameters. The
fit is performed for each grid cell and each month of the year individually. It is further possible to apply the index on different



time scales by defining the number of accumulated months. Similar to (Yu et al., 2023) we choose 6 months, which is within the range of 3-6 months of accumulation for agricultural droughts recommended by the WMO (2012). To quantify the strength of a drought (or wet spell) the probabilities are mapped to SPEI values and can be assigned to categories shown in Table 1 (Guenang and Kamga, 2014). A more detailed guide on the described calculations can be found in McKee et al. (1993) and Guenang and Kamga (2014).

**Table 1.** Mapping of SPEI values to their category and occurrence. These probabilities are used to fit the probability function used by the SPEI (Vicente-Serrano et al., 2010). SPEI values between 0 and -1 for the SPEI have originally been labelled as mild droughts, but are considered as normal for simplification (McKee et al., 1993).

| SPEI | Label | Occurrence |
|---|---|---|
| $2.0 \leq$ SPEI | Extreme wet | 2.3% |
| $1.5 \leq$ SPEI $< 2.0$ | Severe wet | 4.4% |
| $1.0 \leq$ SPEI $< 1.5$ | Moderate wet | 9.2% |
| $-1.0 <$ SPEI $< 1.0$ | Normal | 68.2% |
| $-1.5 <$ SPEI $\leq -1.0$ | Moderate dry | 9.2% |
| $-2.0 <$ SPEI $\leq -1.5$ | Severe dry | 4.4% |
| SPEI $\leq -2.0$ | Extreme dry | 2.3% |

Due to changes in precipitation and potential evapotranspiration in many regions over the last decades, the definition of the reference period is important for the resulting SPEI values. It is recommended to use a reference period of 30 years or more, aligning well with 30 year climate normals which are standard for benchmarking in climate change assessments (WMO, 2017). However, we found that calibrating the SPEI on short reference periods (30 values per month of the year) can lead to an artificial increase in the occurrence of extreme events (Weigel, 2024). Prolonging the period into the past on the other hand decreases the accuracy and availability of global observations of atmospheric variables. As a compromise we choose a reference period of 65 years (1950-2014) and show the evaluation for its last 35 years (1980-2014).

For this study we used the open source SPEI R-package to calculate $ET_0$ and the SPEI (Beguería and Vicente-Serrano, 2023).

## 2.3 Diagnostics

To analyze the drought characteristics, we calculate $ET_0$ and SPEI as described on the same regular 1 ° x 1 ° for 18 CMIP6 models, described in Section 3. We calculate multi-model means of the following metrics to compare them across different scenarios.

As a metric for the non-linear change over time we use a decadal rate for evaluating the model simulations and analyzing projections. The decadal change rate is calculated by subtracting the average of the periods first $N$ years from the last $N$ years



and then normalizing the difference to an average change per 10 years. The result is roughly comparable to linear regressed trends.

$$D_N = \frac{120}{M-N} \left( \sum_{t=M-N}^{M} x_t - \sum_{t=0}^{N} x_t \right) \tag{3}$$

Frequency, severity and duration of events are often used to describe drought characteristics. For large changes in climate, however, it becomes difficult to distinguish individual drought events when they emerge in time or space to large long lasting periods that are considered as a single event. This way a general shift towards drier conditions can result in an artificial decrease of frequency as shown in Xu et al. (2019). Therefore these metrics are not suited to analyze SPEI-based droughts in long-term future projections considering high emission scenarios. As an alternative approach to quantify and visualize the temporal
development of global drought characteristics, we calculate the relative land surface area for certain SPEI ranges including near normal, moderate, severe and extreme droughts and wet spells for each month. The area fraction $A_e$ is calculated from the number of cells in the regular 1° x 1 ° grid weighted by its area $a_{x,y}$ for 18 CMIP6 models.

$$A_e(t) = \frac{1}{A_{land}} \sum_{y=0}^{180} \sum_{x=0}^{360} a_{x,y} \cdot i_{e,x,y}(t) \qquad \text{with } i(t) = \begin{cases} 1, & \text{if SPEI in category} \\ 0, & \text{otherwise} \end{cases} \tag{4}$$

Through the area fraction $A_e$ we can visualize multidimensional SPEI data (time, space, model dimensions) for different
scenarios, while maintaining some temporal and spatial information which would get lost in multi-model temporal or spatial means. This makes the event area fraction a well suited metric to analyze the change of drought characteristics in future projections, especially for comparably rare events like extreme droughts.

To address the change of the SPEI values and their spatial spread in a more detailed way we calculated regional statistics using the IPCC AR6 WG1 reference regions shown in Figure 6 (Iturbide et al., 2020).

To load and pre-process the CMIP6 model simulation output and reanalysis data, described in the next section, ESMValTool is used. ESMValTool is an open source software for community-developed diagnostic and performance metrics to evaluate ESM simulations including CMIP contributions (Eyring et al., 2020; Weigel et al., 2021; Lauer et al., 2020). Since its first release in 2016 (Eyring et al., 2016b), the software has been developed continuously and experienced technical improvements in terms of performance and usability (Righi et al., 2020). The new benchmarking and model evaluation capabilities of ES-
MValTool are shown by Lauer et al. (2024). Existing diagnostics for extreme event analysis are described by Weigel et al. (2021).

The ESMValTool framework contains a dedicated python module, called ESMValCore, for data handling and general processing, and a collection of scientific diagnostics. Recipes can be used to setup experiments as a specification of input data, pre-processing methods and a list of diagnostics to be applied to the data. We used the ESMValCore v2.11 to load the described
data, converted them to comparable physical units, masked and regridded them to the same 1° x 1° regular grid (Andela et al., 2025).



The methods described in this section and multiple plotting routines are added to the ESMValTool framework as a set of diagnostics in the scope of this study. This allows to reproduce the results and re-use the methods for further studies i.e. analysis for the upcoming CMIP7 simulations. The required recipes can be found as `ruhe2024_validation.yml` and

`ruhe2024_scenarios.yml`.

## 3 Data

To asses the severity of agricultural droughts in the future we choose 18 models that participated in the CMIP6 ScenarioMIP and provided simulations of the three Shared Socioeconomic Pathways (SSPs; O'Neill et al., 2016; Meinshausen et al., 2020) that are analyzed in this study:

– SSP1-2.6 as a low greenhouse gas (GHG) emission scenario corresponding to a forcing of 2.6 W m$^{-2}$ and an expected global warming of 1.8 K at the end of the century,

    – SSP2-4.5 as the middle of the road scenario in terms of GHG emissions with an end of the century forcing level of 4.5 W m$^{-2}$ and an expected warming of 2.7 K,

    – SSP5-8.5 as a high GHG emission scenario corresponding to a forcing of 8.5 W m$^{-2}$ in 2100 resulting in an expected

warming of 4.4 K.

The minimum requirement for a model to be included in this study was the availability of all variables required to calculate the reference evapotranspiration (wind speed, surface pressure, total cloud cover, daily minimum and maximum temperature) and precipitation. We limited the selection to only one model per institute, to account each ESMs atmosphere and land component only once. The final selection of models is listed with some additional information in Table 2. Relative humidity is not

considered as a selection requirement because we found that we get similar results by approximating actual vapor pressure from $T_{\min}$.

Throughout this study we also discuss our results with respect to projected soil moisture, which is arguably a drought indicator on its own. Vicente-Serrano et al. (2022) used simulated total column soil moisture as a reference to discuss droughts. It might be a useful indicator for hydrological drought types at long time scales. However, the uncertainties between the

predictions of different CMIP6 models increase with depth and the total depth of all layers is not the same between the models. This makes a meaningful comparison of integrated soil moisture between CMIP models difficult, and also leads to even more uncertainties. Similar discrepancies for CMIP5 are described by Berg et al. (2017). Since the maximum root depths of the biosphere is less than 2 m in most regions of the world (Müller Schmied et al., 2021) and more than 50 % of the roots of most crops are in the upper 20 cm of the soil (Fan et al., 2016), we assume the water content in the upper 10 cm of soil

(surface soil moisture) is a more reliable indicator for agricultural droughts in the context of model simulations than the total column. Another benefit of surface soil moisture is the availability of this quantity from satellite observations and reanalysis to validate models for the historical period. We assumed a density of 1000 kg m$^{-3}$ for converting the surface soil moisture to



**Table 2.** Available CMIP6 models providing variables to calculate $ET_0$ and mrsos

| Dataset | Institute | Atmosphere model | Land model | Reference |
|---|---|---|---|---|
| ACCESS-CM2 | CSIRO-ARCCSS | UKMO UM (GA 7.1) | CABLE2.5 | Dix et al. (2019) |
| AWI-CM-1-1-MR | AWI | | | Semmler et al. (2019) |
| BCC-CSM2-MR | BCC | BCC-AGCM | BCC-AVIM | Xin et al. (2019) |
| CanESM5 | CCCma | CanAM5 | CLASS3.6, CTEM1.2 | Swart et al. (2019) |
| CMCC-ESM2 | CMCC | | CLM4.5-BGC | Lovato et al. (2021) |
| CNRM-CM6-1 | CNRM-CERFACS | | | Voldoire (2019) |
| EC-Earth3-Veg-LR | EC-Earth-Consortium | IFS 36r4 | HTESSEL + LPJ-GUESS | EC-Earth (2020) |
| FGOALS-g3 | CAS | LASG-IAP (GAMIL3) | CAS-LSM | Li (2019) |
| FIO-ESM-2-0 | FIO-QLNM | CAM5 | CLM4 | Song et al. (2019) |
| GFDL-ESM4 | NOAA-GFDL | AM4.1 | LM4.1 | John et al. (2018) |
| GISS-E2-1-G | NASA-GISS | ModelE | ModelE + EntTBM | NASA/GISS (2020) |
| INM-CM5-0 | INM | Inbuilt | Inbuilt | Volodin et al. (2019) |
| IPSL-CM6A-LR | IPSL | LMDZ6A | ORCHIDEEv2 | Boucher et al. (2019) |
| KACE-1-0-G | NIMS-KMA | | | Byun et al. (2019) |
| MIROC6 | MIROC | CCSR-NIES AGCM | MATSIRO | Shiogama et al. (2019) |
| MPI-ESM1-2-LR | MPI-M | ECHAM6.3 | JSBACH3.2 | Schupfner et al. (2021) |
| MRI-ESM2-0 | MRI | MRI-AGCM3.5 + MASINGAR mk-2r4c + MRI-CCM2.1 | HAL | Yukimoto et al. (2019) |
| UKESM1-0-LL | MOHC | | JULES-ES-1.0 | Sellar et al. (2019) |

**Table 3.** Reanalysis and observations used for evaluation of CMIP6 simulations and selecting harvest regions.

| Dataset | Institute | Reference |
|---|---|---|
| ERA5 | ECMWF | Copernicus Climate Change Service et al. (2023), C3S (2023) |
| CRU TS4.07 | CRU | Harris et al. (2020) |
| CDS-SATTELLITE | CDS | Dorigo et al. (2019) |
| GFSAD1KCM | NASA | Teluguntla et al. (2016) |

volumetric percent. This way surface soil moisture values from the CMIP6 model simulations are comparable to the reanalysis and observations listed below.

To evaluate the model data we choose ECMWF's monthly averaged ERA5 reanalysis as primary reference (Hersbach et al., 2020). The successor of the former commonly used ERA-Interim dataset does not provide true measurements, but a wide range of variables and global coverage. It is based on the Integrated Forecasting System (IFS) Cy41r2 operating on a 31km spatial and hourly temporal resolution on 137 vertical levels. In the ERA5 reanalysis billions of observations including data



from over 200 satellite mounted instruments, in situ observations, balloons and aircraft measurements (Hersbach et al., 2020)
are assimilated. Full spatial and temporal coverage enables direct comparison with CMIP6 model output. Most variables are
available in the *ERA5 monthly averaged data on single levels from 1940 to present* dataset provided by the CDS (Copernicus
Climate Change Service et al., 2023). Daily maximum and minimum temperatures have been derived from *ERA5 hourly data
on single levels* (C3S, 2023) using the CDS toolbox[1].

In addition to the ERA5 dataset we use the Climatic Research Unit (CRU) gridded Time Series version 4.07 (TS4.07) as a
sanity check for the $ET_0$ calculation method and as alternative reference for temperature and precipitation in the evaluation
process (Harris et al., 2020). In addition we used the CDS's dataset for soil moisture from combined satellite data (CDS-
SM; Dorigo et al., 2019). The dataset is blended from a set of passive (starting 1978) and active (starting 1991) microwave
measurements with a resolution of 0.25 °. A detailed list of used sensors can be found in the datasets user guide Preimesberger
et al..

## 4 Results

In the following we evaluate the last 35 years (1980-2014) of historical simulations by 18 CMIP6 models with ERA5 as
primary reanalysis dataset. Characteristics of agricultural droughts in future projections (2015-2100) by the same models for
three different scenarios are analyzed and discussed in Section 4.2.

### 4.1 Evaluation of historical Simulations

We are interested in the capabilities of models to project long-term characteristics like mean and change rates of drought
relevant variables. Therefore we compare these variables with observations and reanalysis with time integrated metrics for the
period 1950-2014 to estimate the reliability of these simulated variables. The pattern correlations of all 18 analyzed models,
CRU TS v. 4.07 (Harris et al., 2020) and CDS-SM data (Dorigo et al., 2019) with ERA5 (Copernicus Climate Change Service
et al., 2023) are shown in Figure 1.

The temporal averaged pattern of CMIP6 models over the period 1980-2014 show high similarity ($P$>0.8 for most models)
with the ERA5 reference data for surface downwelling shortwave radiation ($RS_\text{down}$), reference evapotranspiration ($ET_0$),
surface pressure ($p_\text{surf}$) and daily minimum and maximum temperature ($T_\text{min}$, $T_\text{max}$). Lower pattern correlation (0.6<$P$<0.8) is
found for precipitation ($PR$). The agreement of wind speed at 10 m ($V_\text{10m}$) and soil moisture ($SM_\text{surf}$) patterns is the lowest
with correlation coefficients $P$<0.6 for most models, also with a higher spread between the models. The averaged soil moisture
pattern of the CDS-SM dataset based on satellite observation does not show higher correlation to ERA5 than the models, which
makes it difficult to judge model performance based solely on ERA5. However, we see high agreements in the most important
input variables for $ET_0$ ($RS_\text{down}$, $p_\text{surf}$, $T_\text{min}$ and $T_\text{max}$). This propagates to a $ET_0$ pattern correlation above 0.8 for 12 out of 18
analyzed models.

---

[1]Copernicus dropped support of the Climate Data Stores toolbox feature. As an alternative the variables *minimum-* and *maxi-*
*mum_2m_temperature_since_previous_post_processing* can be downloaded from ERA5 daily data on single levels dataset and averaged monthly.





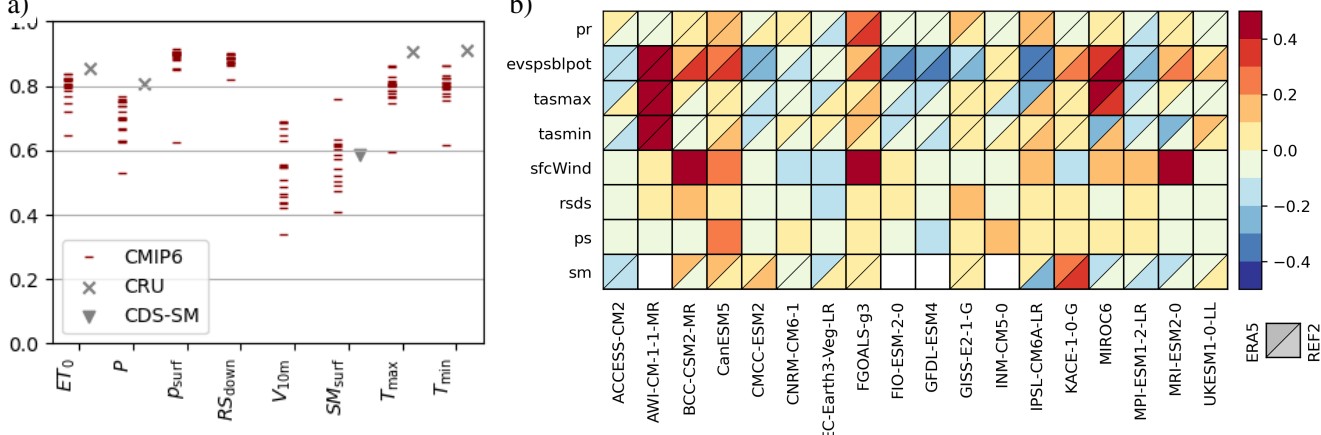

**Figure 1.** Overview of the CMIP6 multi-model performance compared to ERA5 reanalysis over land surface. In panel a pattern correlation of historical CMIP6 simulations with ERA5 for precipitation ($PR$), reference evapotranspiration $ET_0$, daily minimum and maximum temperature ($T_{min}$, $T_{max}$), wind speed at 10 m ($V_{10m}$), surface down welling shortwave radiation ($RS_{down}$), surface pressure ($p_{surf}$) and soil moisture ($SM_{surf}$) are shown. The data is averaged over the period 1980 to 2014. The additional markers show pattern correlation of ERA5 to CRU and CDS-SM. In panel b the relative model performance for the same variables is shown. The root mean squared distance (RMSD) of each model to ERA5 is centered and normalized at the median RMSD. Negative (blue) values indicate that the model predicts results more similar with ERA5 compared to the other models. Red colors indicate worse performance. For some variables the performance compared to CRU is shown as well in the bottom right triangle. The alternative reference for $SM_{surf}$ is CDS-SM.

Figure 1b shows the centered median between each models root mean squared distance (RMSD) when compared with ERA5 (upper left rectangles). For the models providing soil moisture the CDS-SM dataset is used as an alternative reference REF2 (bottom right). For the other variables CRU is used as alternative, where available. Due to the normalization, this plot lacks information about the actual distance to the reference, but instead highlights the relative performance between the models for multiple variables. A blue color indicates smaller errors to the reference and therefore better performance compared to the other models. Errors larger than multi-model median are shown in red. Biases in model simulations directly contribute to RMSD, while the normalized SPEI is able to ignore them. Therefore, it is possible that models with high RMSD are still able to project drought conditions reliably. The AWI-CM-1-1-MR and MIROC6 models for example show the highest RMSD to ERA5 for $ET_0$ and $T_{min}$. BCC-CSM2-MR, CanESM5 and FGOALS-g3 simulate $ET_0$ more similar to ERA5 but with large differences to the alternative reference dataset CRU. Throughout all models ACCESS-CM2, CMCC-ESM2, CNRM-CM6-1, EC-Earth3-Veg-LR, FIO-ESM-2-0, GFDL-ESM4, and MPI-ESM1-2-LR perform better in simulating $ET_0$ than the ensemble median in respect to both reference datasets. For soil moisture we find low agreement in RMSDs between reference datasets ERA5 and the CDS-SM satellite observations. We are not able to comment on model performance in these cases.

In the IPCC AR6 a similar method has been used to evaluate CMIP5 and CMIP6 model performance for the period 1980–1999 (Eyring et al., 2021, Fig. 3.42). Comparing precipitation, where ERA5 has been used for reference, we find that our





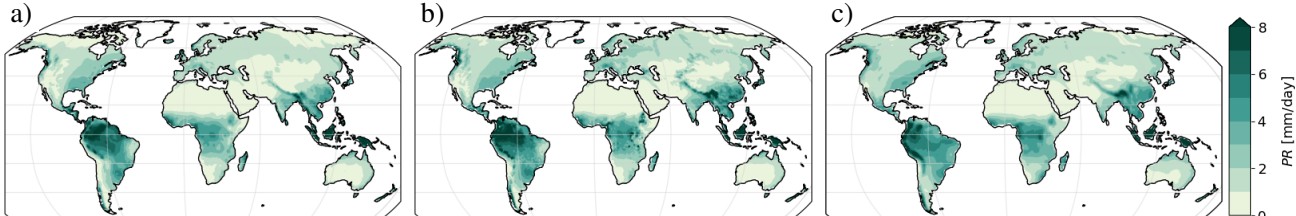

**Figure 2.** Global pattern of temporal mean precipitation based on a) ERA5, b) CRU and c) CMIP6 multi-model mean over the period 1980-2014.

centralized RMSD values are generally higher, because the median model in our subset is performing better for precipitation than the median of the 58 model ensemble analyzed in Eyring et al. (2021). Despite a few exceptions, the ranking of the models is the same with a few exceptions.

Due to the importance of precipitation and reference evapotranspiration for drought formation, they are evaluated further in the following. Similar figures for other variables can be found in the appendix (A1, A2, A3).

Precipitation is the main source of fresh water for most agricultural regions and therefore a crucial variable in drought analysis. In the multi-model mean of CMPI6 models and both reanalysis datasets we find mean patterns that qualitatively agree over the period 1980-2014 in Figure 2. Average daily precipitation close to 0 mm can be found for dry regions in northern Africa and South-west Asia. Regions of high precipitation caused by topological boundaries like mountain ranges can be seen clearly for the Andes in South America and for the Himalayas, which forms the northern boundary of the Indian monsoon climate in the CMIP6 multi-model mean and both reference datasets. The CMIP6 simulations also agree with ERA5 and CRU reanalysis precipitation over rain forest regions of South America (8-10 mm per day) and Central Africa (3-6 mm per day).





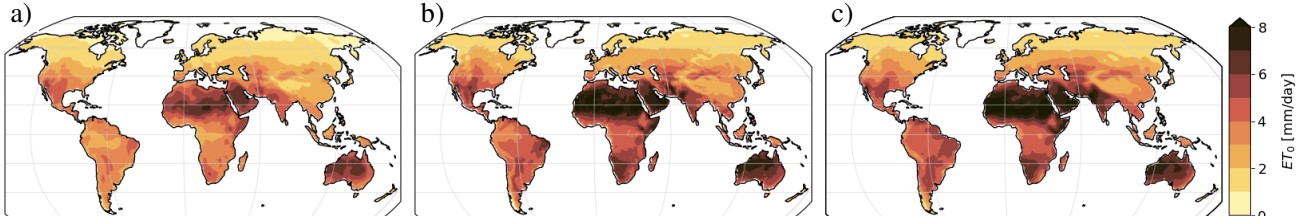

**Figure 3.** Global pattern of temporal mean of reference evapotranspiration ($ET_0$) based on a) ERA5, b) CRU and c) CMIP6 multi-model mean over the period 1980-2014. For CRU the variable is taken from the dataset. For ERA5 and CMIP6 $ET_0$ is calculated from other atmospheric variables (Eq. 1) similar to the method used for CRU.

$ET_0$ is calculated for ERA5 and the CMIP6 model output using Equation 1 based on the same set of variables. The $ET_0$ provided by the CRU dataset is derived using a similar method, but with a crop height of 0.12 m and included relative humidity to calculate the vapor pressure deficit (Harris et al., 2020; Ekström et al., 2007). The differences in the methods might cause some systematical differences, but we can still see a general agreement in the $ET_0$ pattern over the period 1950-2014 between all datasets in Figure 3. Further, we can find that models reproduce expected temporal mean features like decreasing $ET_0$ towards high latitudes due to decreasing temperatures, low $ET_0$ at the high altitude Tibetan Plateau and highest $ET_0$ values at the Sahara due to high temperature and low cloud coverage in an arid region.

## 4.2 Projections

For this study only non glaciated land area is considered and referred to as global in the following. The global mean of $PR$ and $ET_0$ for the base period and different future scenarios are shown in Figure 4. For both variables a higher positive trend can be noticed with higher emission scenarios from SSP1-2.6 to SSP5-8.5. The increase of $ET_0$ is significantly higher compared to precipitation. This results in a decreasing water budget for future projections and a general shift of SPEI towards drier conditions, which we can see in the corresponding annual SPEI time series, the third plot in Fig. 4. All 18 analyzed models agree in a projected decrease in global mean water budget over the period 2015-2100 in SSP2-4.5 and SSP5-8.5 scenarios. The general downward trend of water budget and a shift towards drier climate in higher emission scenarios agrees with several other studies (Balting et al., 2021; Vicente-Serrano et al., 2022).

However, these changes are not distributed uniformly over the globe. We found significant regional differences in quantity of water budget and therefore SPEI changes in future scenarios as shown in Figure 5. It shows maps of the multi-model mean decadal change of SPEI over the period 2015-2100 for three different future scenarios. The SPEI is calculated and calibrated for each model individually using 1950-2014 as common baseline for all three scenarios. While the change rates are different between the scenarios, the general pattern and signs of the trends are the same for most regions. Beside the decrease in most parts of North and South America, Africa and Southwest Asia small regions with increasing water budget can be identified, mainly in the Arctic and Subarctic as well as some mountain regions with relatively low crop production. Positive and negative changes are intensified for the SSP5-8.5 scenario compared to SSP1-2.6.



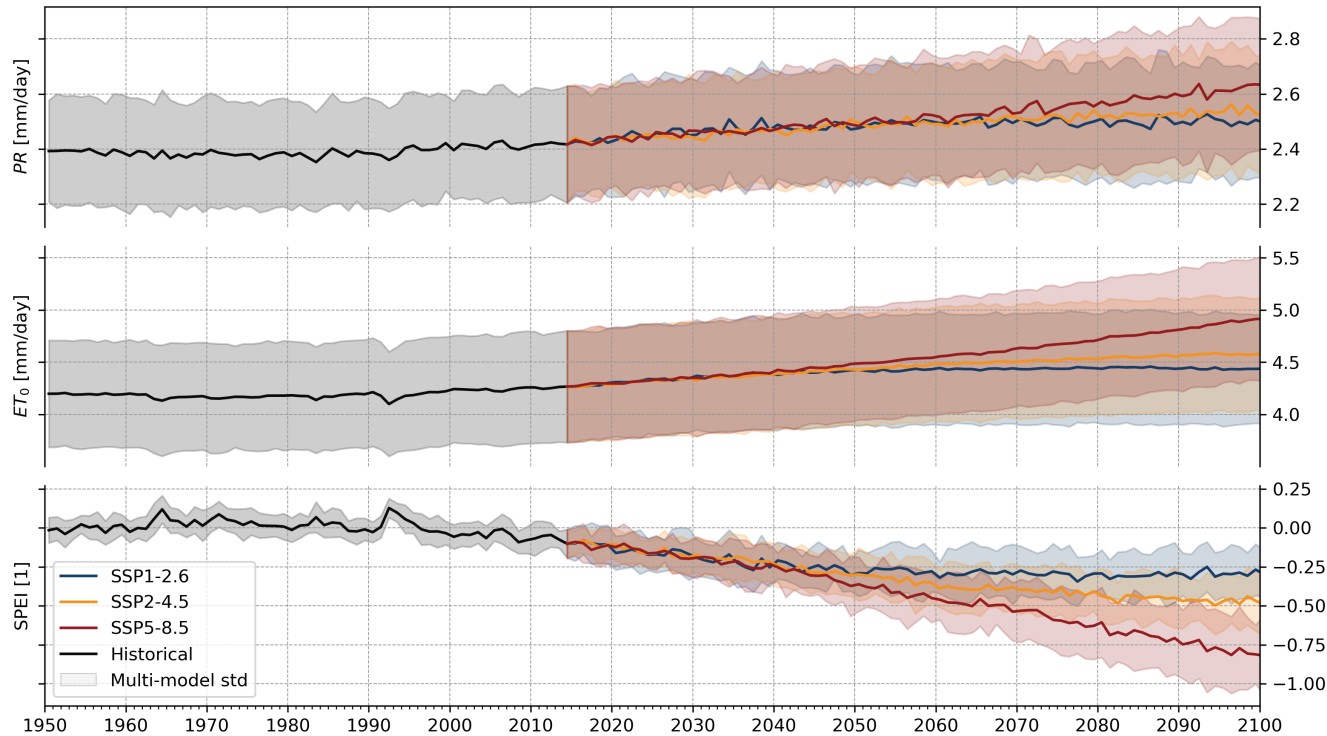

**Figure 4.** Annual precipitation, reference evapotranspiration and the derived SPEI. All variables are averaged over global land surface. The multi-model means of 18 CMIP6 models are drawn as solid lines, with shaded areas indicating the multi-model standard deviation. For the period (2015-2100) different future scenarios can be distinguished by color.

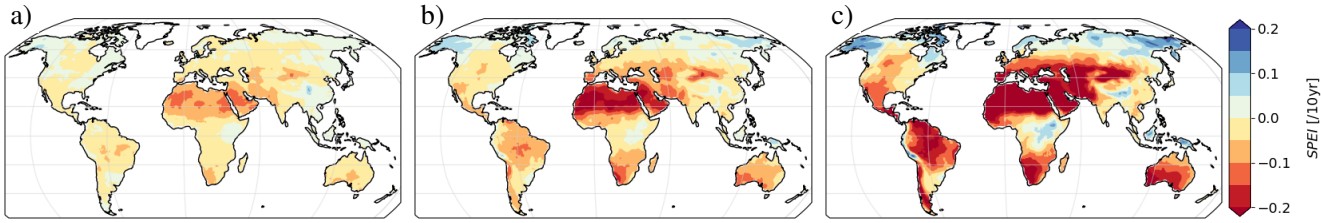

**Figure 5.** Decadal change rate of SPEI with reference period 1950-2014 derived from CMIP6 projections for different future scenarios. The change is defined as the absolute difference between 2015-2034 and 2081-2100 normalized to the average change over 10 years.

We further made use of the IPCC AR6 WG1 Land reference regions (Iturbide et al., 2020) to apply regionally integrated statistics. Our main focus are regions with a significant amount of crop production and agricultural land use. We identified these regions by their relative amount of harvest area in 2010 according to the GFSAD1KCM dataset (Teluguntla et al., 2016). Figure 6 shows a regridded combination of major and minor rain-fed and irrigated harvest area masks. The reference regions are shown as overlay in panel a). Panel b) shows the relative amount of harvest area for each region. Regions above 33.3%





a)

b)

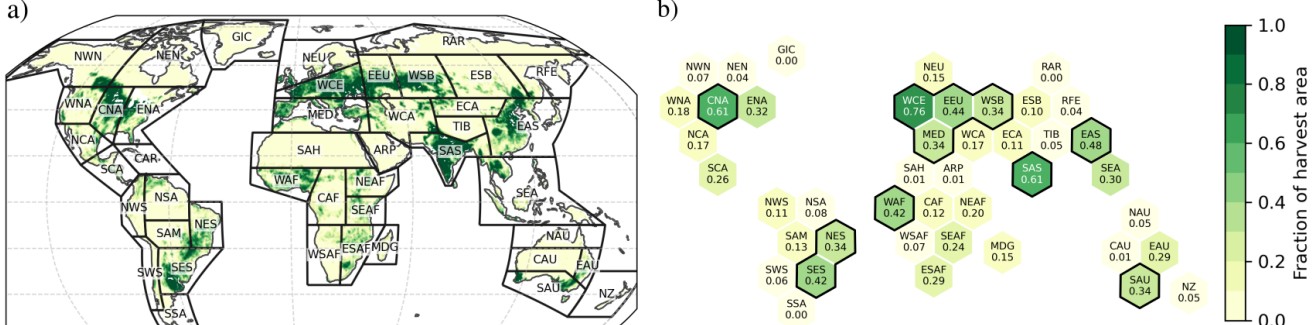

**Figure 6.** Selected regions with at least 33% harvest area. In panel a) the land subset of the IPCC AR6 WG1 reference regions is shown as overlay(Iturbide et al., 2020) . The color represents the regridded relative harvest area derived from a combination of irrigated and rain fed masks in GFSAD1KMCM (Teluguntla et al., 2016).

harvest area are considered as highly relevant for agriculture throughout this study. The agricultural most relevant regions are namely Western&Central-Europe (WCE), C.North-America (CNA), S.Asia (SAS), E.Asia (EAS), E.Europe (EEU), Western-Africa (WAF), S.E.South-America (SES), W.Siberia (WSB), Mediterranean (MED), N.E.South-America (NES), S.Australia (SAU). They are highlighted in 6b.




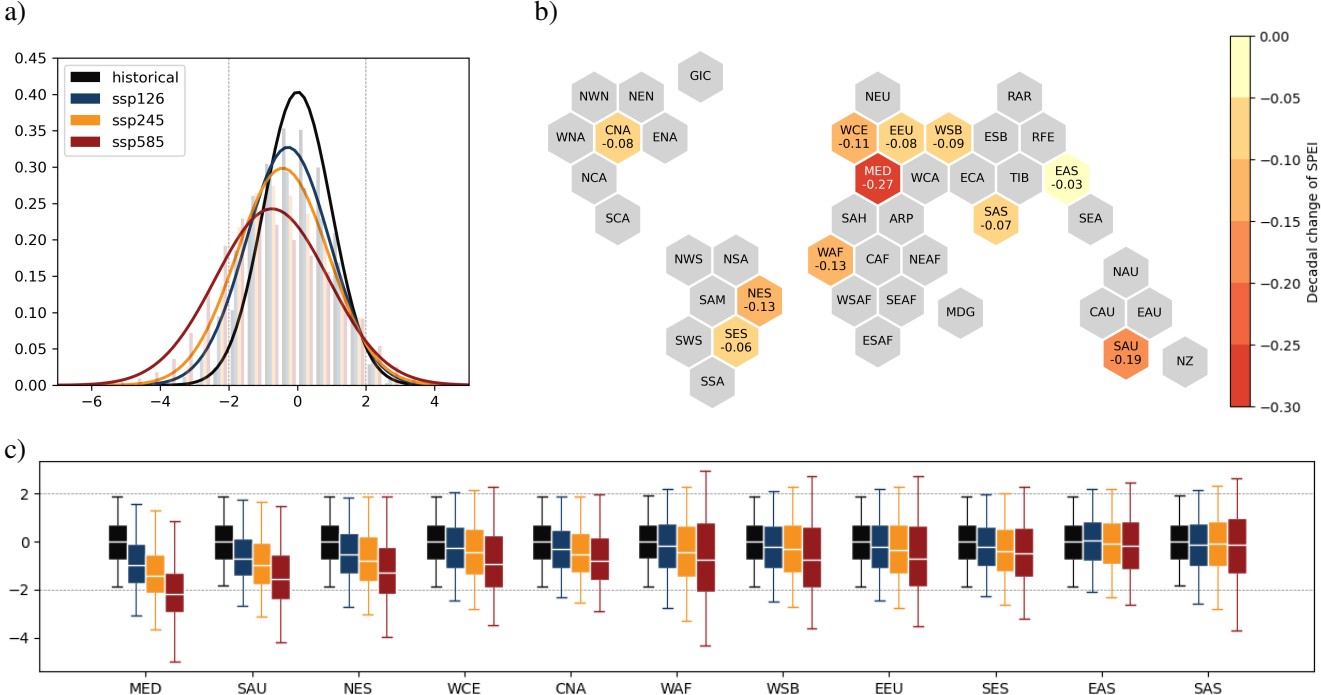

**Figure 7.** SPEI distributions for the CMIP6 multi-model ensemble in different future scenarios. The SPEI distributions for all grid cells in selected harvest regions are shown in panel a) and regional distributions in form of boxplots for IPCC AR6 WG1 reference regions (Antarctic excluded) in panel c). The hexagons in panel b) show the mean decadal change of mean SPEI for each of the selected region but only for the SSP5-8.5 scenario. The regions are positioned relative to each other based on their real locations.

Change rates on their own are not sufficient to describe the development of projected drought characteristics. Due to regional
differences we analyze the statistical distribution of the drought index for individual regions of similar climatic conditions. An
overview of the distribution for combined and individual agricultural relevant regions is given in Figure 7. The same figure for
global distribution and all non glaciated land regions is appended as A6 and for semi arid regions of Europe and Asia as A8.

In Figure 7a we can see the distribution of index values for the reference period 1950-2014 and the last 30 years of the
projected century 2070-2100 for three future scenarios, collected over the most relevant agricultural regions. While the values
in the reference period follow a Gaussian distribution, which is centered at zero by definition, the distributions of the future
scenarios are shifted towards negative indices (drier climate), which agrees with the observed global downward trend in Figure
4. We also expect more extremes for higher emission scenarios, according to widening distributions.

The hexagons in panel b) are shown to provide a spatial overview of the regions and their projected SPEI decadal change
in the SSP5-8.5 scenario. Due to the symmetric Gaussian distribution of the SPEI the decadal change of the mean is roughly
proportional to the shift of the distribution. Similar to Figure 5 the lower emission scenarios SSP1-2.6 and SSP2-4.5 show
smaller change rates (Figure A5) but qualitatively agree with Figure 7b for most regions.



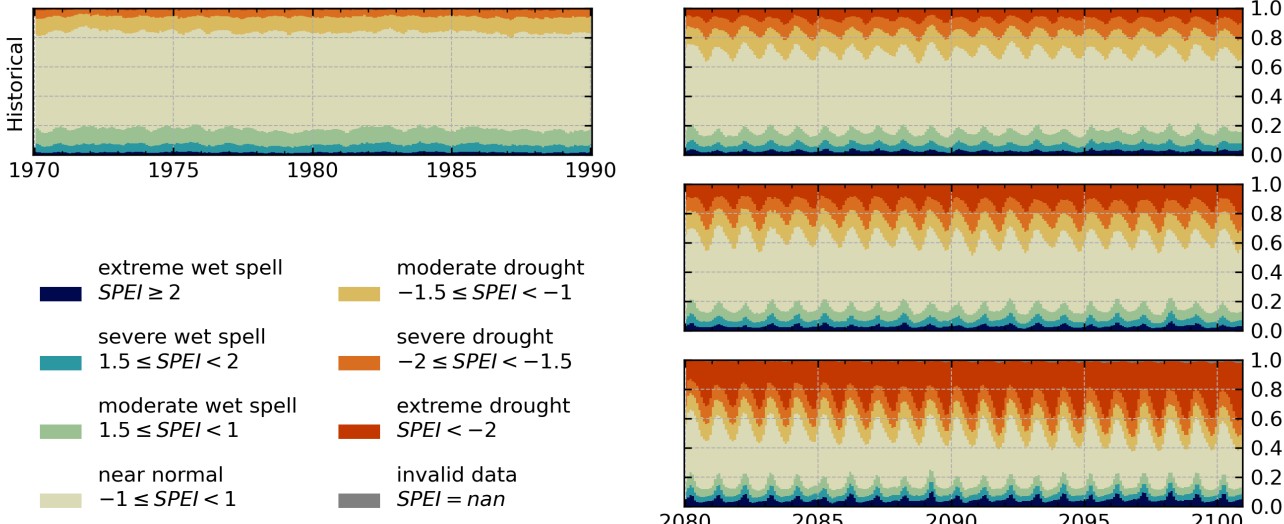

**Figure 8.** Timeseries of relative drought event area according to SPEI. The color indicates how much of the agricultural relevant regions area is covered by certain drought conditions for each month. A 20 year interval from the baseline period 1950-2014 is shown on the left, while the plots on the right display relative area by drought category over the last 20 years (2080-2100) of different future scenarios.

The regional statistics in Figure 7c show the distributions as box plots for the IPCC AR6 WG1 land reference regions, ordered by median in the SSP5-8.5 scenario. For nine of the 11 regions a clear shift towards lower median SPEI values enhancing with higher emission scenarios can be found. This implies a general shift towards drier conditions and we find more severe and extreme droughts towards the end of the century in all three future scenarios based on SPEI projections of 18 ESMs. For the MED region we see the median of the drought index for 2070-2100 being below -2 following the SSP5-8.5. Which means the 2.3% driest conditions during the simulated period 1950-2014 can be considered as normal in the Mediterranean during the projected period 2070-2100. The Mediterranean is mentioned as a hotspot for increase of extreme events including heat waves and droughts in other studies (Balting et al., 2021; Paçal et al., 2023). For East and South Asia no general shift can be found, but the number of months classified as extreme droughts is also higher compared to historical reference.

The first five of all global non-glaciated land regions (Figure A6) SAH, ARP, MED, WCA and ECA are neighbors and part of northern Africa, southern Europe and western Asia. A general shift towards drier conditions with increasing forcing scenarios can be seen for 32 of the 42 regions. Seven regions (GIC, RAR, RFE, NEN, NWN, NEU, CAF) show an opposite shift towards wetter conditions. Six of them are are located at high latitudes. While the widening of the index distribution for future scenarios can be found for all of the analyzed regions, it is most prominent in Western Africa (WAF) region.

Using the ability of the SPEI to assign a category for the severity of drought based on climatic conditions we calculated the area that is covered by drought conditions of a certain category at any time. This allows us to quantify the area that might be affected by droughts under changing atmospheric conditions in future projections.





In contrast to spacial mean time series the area fraction plots show the increase in wet and dry conditions without wet and dry events compensating each other. Further, they are not primarily described by the impact of certain regions where the average SPEI tend to increase beyond the threshold for extreme events. Figure 8 compares the event area fraction for an intervals during the historical period (1970-1990) with the end of century (2070-2100) over the selected harvest area. The complete series (1950-2100) is appended as Figure A7.

The first interval in Figure 8 presents the relative surface area for each index category during the calibration period. As each individual grid cell and month of the year is calibrated to match a normal distribution with certain shape, it is expected that we see 2.3 % of the area under extreme drought conditions without seasonal cycle on average. The interval 1970-1990 roughly represents this. However, the 2080-2100 interval selected from the end of the projected future shows significantly more area under moderate to extreme droughts for all three scenarios especially in autumn. Further the temporal resolution reveals an increasing seasonal dependency of the spatial extend of droughts. According to the SPEI for future projections following a worst case scenario SSP5-8.5 between 20 and 40 % of the land area in agricultural active regions would experience extreme drought conditions at the end of century. In contrast, for the SSP1-2.6 the extreme drought area does not exceeds 10 %. For the drought categories we find peaks in northern hemisphere autumn (September-November), resulting from high evapotranspiration and low precipitation accumulated over the previous six months in spring and summer. In the extreme drought category we identify the largest differences between the scenarios. However, the peak area under at least severe (moderate) conditions also ranges from 20 % (35 %) to 50 % (60 %) throughout the scenarios.

For wet spells we find a similar increasing seasonality with opposite phase peaking in early spring. The area of wet spells with SPEI > 1.5 does not clearly expand from the expected occurrence rate of 16.9 % (Table 1). It stays below 20 % on yearly multi-model average in all scenarios. Nevertheless we see a significant growth in area of extreme wet spells, which is consistent with the broadening and shifting of the global SPEI distribution for different scenarios (Figure 7a), which overlap at SPEI between -1 and -2.

### 4.3 Discussion

The historical evaluation in 4.1, was meant to be a sanity check for the $ET_0$ calculation methods and a validation for the model projections in the first place. Considering the spread of $ET_0$ and $P$ between CMIP6 model projections some sort of performance based weighting in the multi model ensemble could improve the results based on a more sophisticated evaluation taking seasonal distributions and biases into account. The differences in reanalysis and lack of reliable global observations especially for wind and precipitation is an additional challenge Weigel (2024) Soci et al. (2024).

We decided to use the SPEI for our analysis of the development of agricultural drought characteristics, due to its ability to be applied globally while being spatially comparable over different regions. However, the statistical approach of standardized indices implies that the intensity or severity of a drought is calculated based on its rarity and not based on the actual physical soil parameters or how it impacts crop growths. Analyzing the actual water balance levels and how they impact yield for different crop types on a regional level would be a valuable addition to this study.




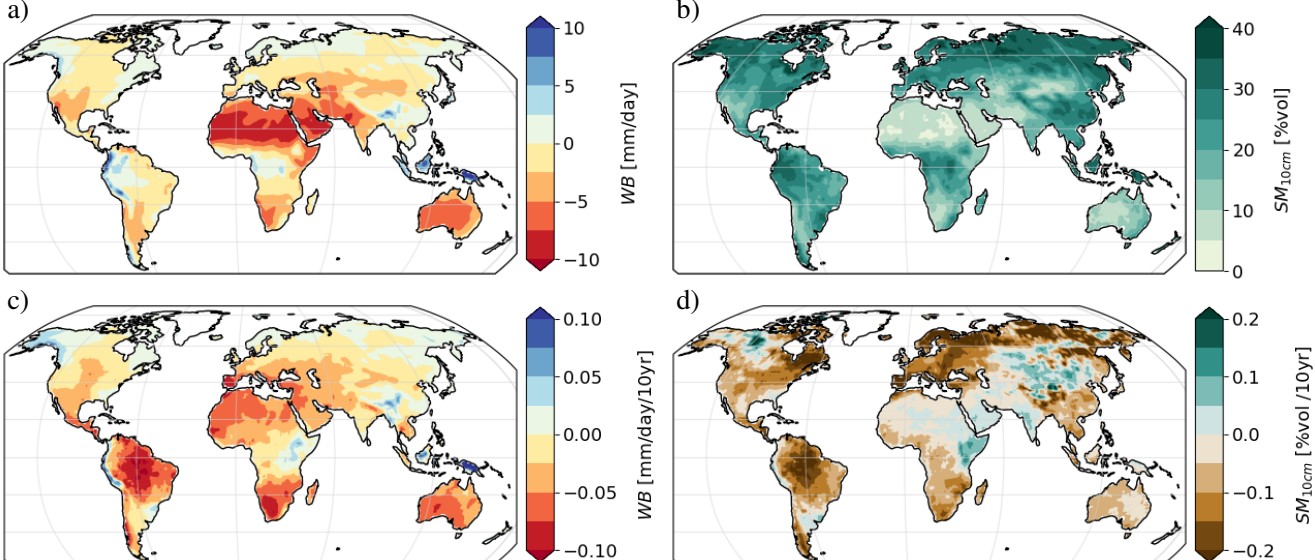

**Figure 9.** Temporal average a,b) and decadal change c,d) in a,c) global water budget ($PR$-$ET_0$) and b,d) soil moisture projected from CMIP6 multi-model ensemble for the SSP2.4-5 scenario (2015-2100).

Beside the relative nature of the index we also have to consider that most of the area with highest increase in droughts in high emission scenarios are already arid regions i.e. the Sahara desert. However, Singh et al. (2022) found a tenfold higher agricultural exposure to severe compound droughts towards the end of century using model projections from RCP8.5 in CMIP5.

While root zone soil moisture is a good indicator for agricultural droughts, the availability of reliable global soil moisture data is limited and simulations show high inter model variability (Figure 1). Further, available water is not the only parameter responsible for crop failure. A high atmospheric evaporative demand can also cause increasing plant water stress. Figure 9 illustrates the difference between a purely hydrological approach to quantify droughts like soil moisture and approaches like SPEI, that consider the influence of $ET_0$. In the upper panels (a,b), we can see a that regions with an average water budget

below -5 mm per day correspond to arid regions with low soil moisture. Especially Northern Africa, South West Asia and Australia. However, under the SSP2-4.5 future scenario, soil moisture is expected to decrease over Europe and South America, but shows no or even positive changes in arid regions (Figure 9d), because there is not much water available to evaporate or runoff (other scenarios can be found in A4). The atmospheric demand of water, shown in Figure 9c, can further increase even if the soil does not contain water at all. However, the influence of $ET_0$ on crop failure and agricultural drought severity might

need further research to be quantified exactly.

Due to the above mentioned reasons the SPEI might be suited to investigate the impact of changes in climate to the characteristics of droughts, but this is not directly transferable to any ecological impact regardless of the focuses on regions with lot of harvest area. With this study we hope to provide further motivation for research on regional impact. Balting et al. (2021) analyzed SPEI characteristics projected by CMIP6 models for the same three future scenarios, with a different set of models





and focus on the northern hemisphere in summer (JJA). We do not show occurrence rates in this work, but based on regional distribution shifts and the increase of extreme dry area in we can confirm the highest increase in projected droughts for the MED region, less but still increases in WCE EEU and WSB. By investigating the change of extreme drought (SPEI < -2) in addition to severe droughts, we find higher difference between the future scenarios for regions where the SPEI stays below -1.5 for many months. We also complement their study by providing SPEI analysis for additional regions and over all seasons, even

though it is most important for agriculture during growing season.

Although, the SPEI is symmetric, which makes it possible to apply it to wet spells in a similar way, it is not well suited to identify extreme events (i.e. floods). Therefore, we focus the analysis on droughts, while still discussing the whole distribution of the index over all categories.

## 5 Conclusions

To evaluate the performance of 18 CMIP6 models for simulation of drought related variables, we compare their simulations of precipitation $PR$, surface wind speed $V_{10m}$, downwelling solar radiation $RS_{down}$, surface pressure $p_{surf}$ and daily minimum and maximum temperature $T_{min}$ $T_{max}$ monthly averaged for the last 65 years of the historical experiment (1950-2014). As reference for the evaluation we use the monthly and hourly ERA5 reanalysis datasets (Copernicus Climate Change Service et al., 2023; C3S, 2023). We find generally high agreement between reanalysis and all CMIP6 models for $T_{min}$, $T_{max}$ $p_{surf}$ and $RS_{down}$.

They show similar global trends and temporal mean pattern correlations of 0.8 and higher. In contrast precipitation shows low agreement between different models and reanalysis while the trends in several regions are significant and directly impact SPEI. Since we found that CRU and ERA5 show very different precipitation patterns for that time period we are not rating the performance of individual models and rather incorporate the knowledge that precipitation comes with huge uncertainties in reanalysis and simulated variables in our analysis. Similar to the study by Vicente-Serrano et al. (2022), we find that uncertainty

in precipitation data limits the quality of global drought assessment. The approximated $ET_0$ values based on CMIP6 model predictions, are considered to be reliable even though, their significance depend on the operational definition of a drought. For the 6-month SPEI we identified $ET_0$ as the main driver of decrease in water budget for many regions.

In agreement with the results of Balting et al. (2021) we find negative trends of water budget in the Mediterranean and several other regions. The trends are significantly stronger towards dry climate in scenarios of higher warming levels. The drought

conditions that have been considered as the driest 2.3 % of each month 1950-2014 become the new normal in several arid regions at the end of this century following the SSP5-8.5. In this scenarios last two decades the projected SPEI shows extreme drought conditions with peak spatial extend of 40 % of the land surface in agricultural relevant regions. With an estimated temperature increase of 2.7 degrees at the end of century following the SSP2-4.5 pathway we calculated area fraction of maximum 20 %, and maximum 10 % in the SSP1-2.6 projection experiment for an estimated temperature increase of 2 degrees.

The increasing $ET_0$ is the dominant driver for drier conditions in arid and semi arid regions. Even with high uncertainties in precipitation projections it can be shown that the number of extreme droughts globally increases in SSP2-4.5 and SSP5-8.5, with seasonal peaks in autumn.





Relative indices such as the SPEI require a baseline period as reference. The length and the starting point of this period has an impact on the resulting Index values. Therefore, the results always need to be interpreted with respect to the reference

period and are not directly comparable between analysis with different reference periods. The SPEI maps changes in the water budget to strengths of droughts only by taking their distribution in the past into account. For regions with low variance extreme conditions can be found for small water budget changes, even if crops would be capable of adapting to such small changes.

We found significant higher global increase in moderate and extreme atmospheric drying conditions on a six months timescale, in higher emission future scenarios. This also holds for most agricultural relevant IPCC WG1 reference regions.



*Code and data availability.* The code to reproduce this study is part of ESMValTool v2 (Righi et al., 2020; Eyring et al., 2020). The corresponding recipes can be found under *droughts/lindenlaub25*. ESMValTool v2 is released under the Apache License, version 2.0 and its latest release is publicly available on Zenodo at https://doi.org/10.5281/zenodo.3401363 (Andela et al., 2024). The source code of the ESMVal-Core package, which is installed as a dependency of ESMValTool v2, is used to preprocess the data and also publicly available on Zenodo at https://doi.org/10.5281/zenodo.3387139 (Andela et al., 2025). ESMValTool and ESMValCore are developed on the GitHub repositories available at https://github.com/ESMValGroup (last access: 18 March 2025).

*Data availability.* CMIP6 data are freely and publicly available from the Earth System Grid Federation (ESGF) and can be retrieved by ESMValTool automatically by setting the configuration option *search_esgf*. All observations and reanalysis data used are described in 3. The observational and reanalysis datasets are not distributed with ESMValTool, which is restricted to the code as open-source software, but ESMValTool provides a collection of scripts with downloading and processing instructions to recreate observational and reanalysis datasets used in this publication.

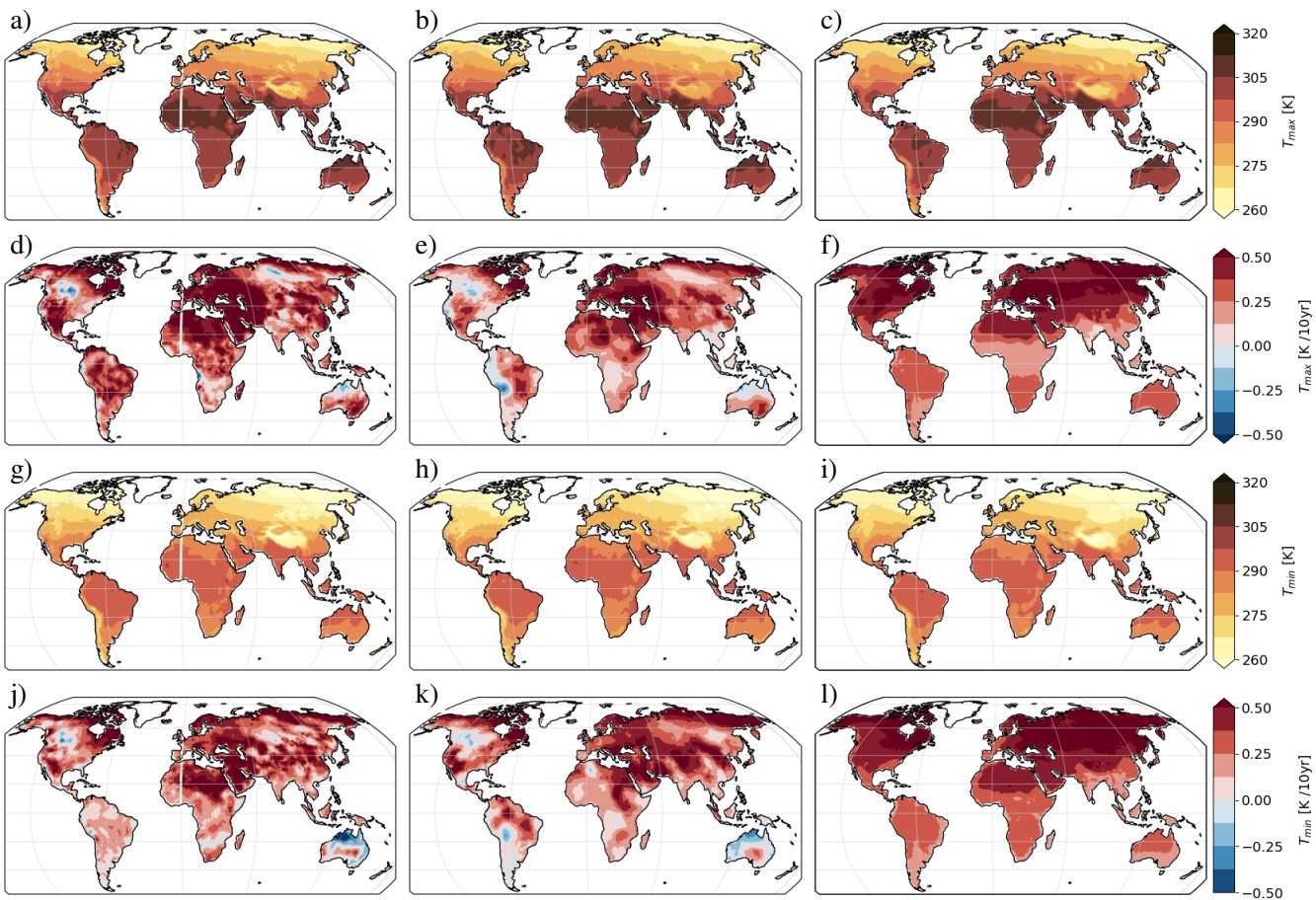

**Figure A1.** Global pattern of temporal mean (a,b,c) and average change (d,e,f) of temperature at 2m height based on ERA5 (a,d), CRU (b,e) and CMIP6 multi-model mean (c,f) over the period 1950-2014.





**Figure A2.** Global pattern of temporal mean (a,b,e,f) and average decadal change (c,d,g,h) of surface pressure $p_{sfc}$ (a-d) and wind speed at 10 m (e-g) based on ERA5-Land (a,c,e,g) and CMIP6 multi-model mean (b,d,f) over the period 1950-2014.



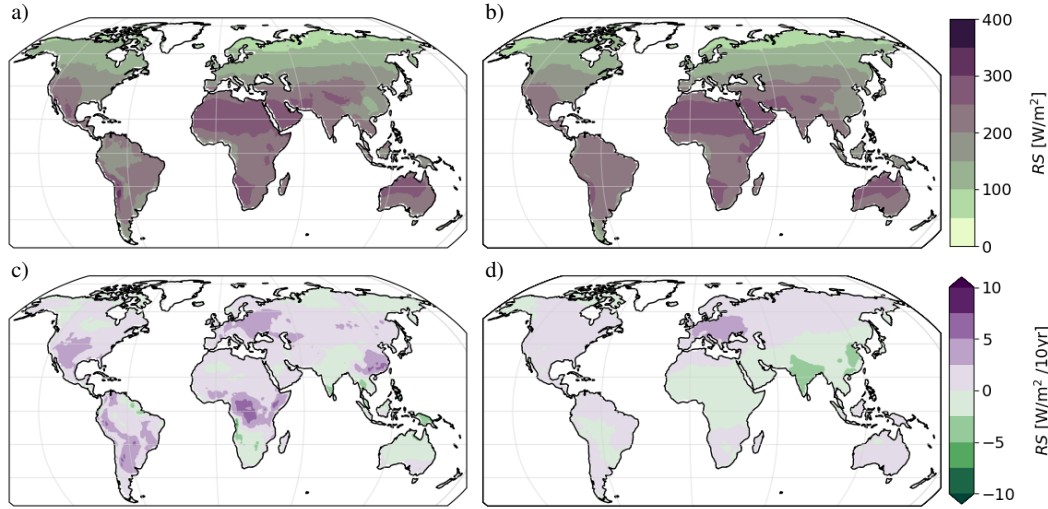

**Figure A3.** Global pattern of temporal mean (a,b) and average change (c,d) of downward surface solar radiation based on ERA5 and CMIP6 multi-model mean over the period 1979-2014.



**Figure A4.** Change rate and average of soil moisture (a-f) and water budget (g-l) in different future scenarios (a,d,g,j: SSP 1-2.6, b,e,h,k: SSP 2-4.5, c,f,i,l: SSP 5-8.5) from 2015-2100.



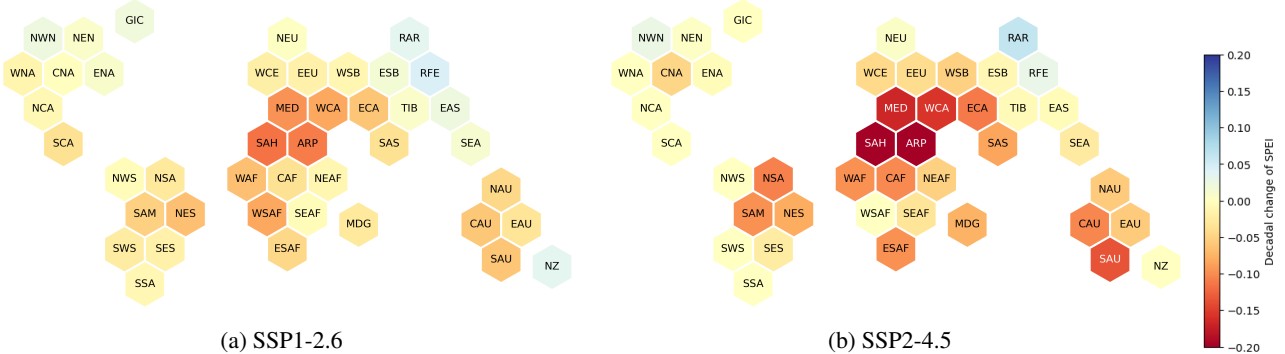

**Figure A5.** Average decadal change between 2015-2034 and 2081-2100 of SPEI in different future scenarios. Similar plot for SSP5-8.5 is shown in Figure 7b.

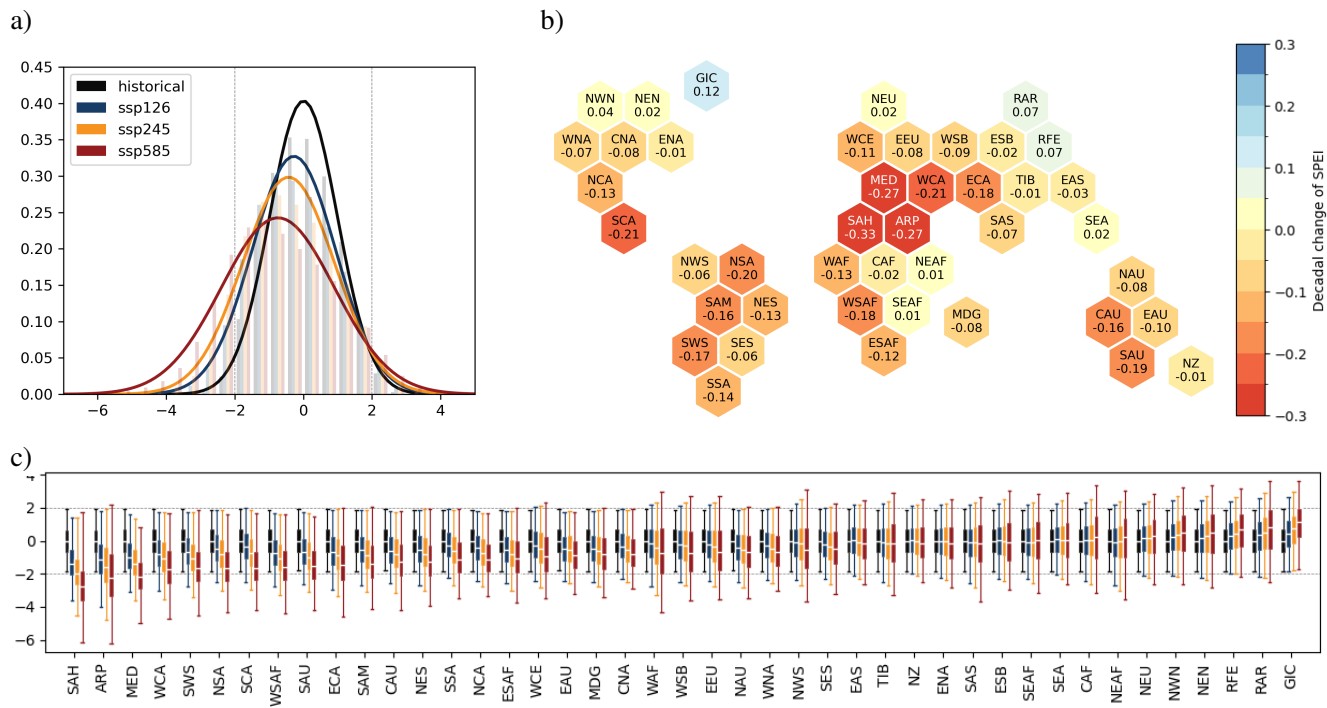

**Figure A6.** Global overview of SPEI values for the CMIP6 multi-model ensemble for different future scenarios. The global SPEI distributions for all non glaciated land grid cells for the historical period (1950-2014) and end of century (2070-2100) are shown in panel a) and regional distributions for the same periods in form of boxplots for IPCC AR6 WG1 reference regions in panel c). The hexagons in panel b) show the regional mean average decadal change of SPEI for each region only for SSP5-8.5. The regions are positioned relative to each other based on their real locations.



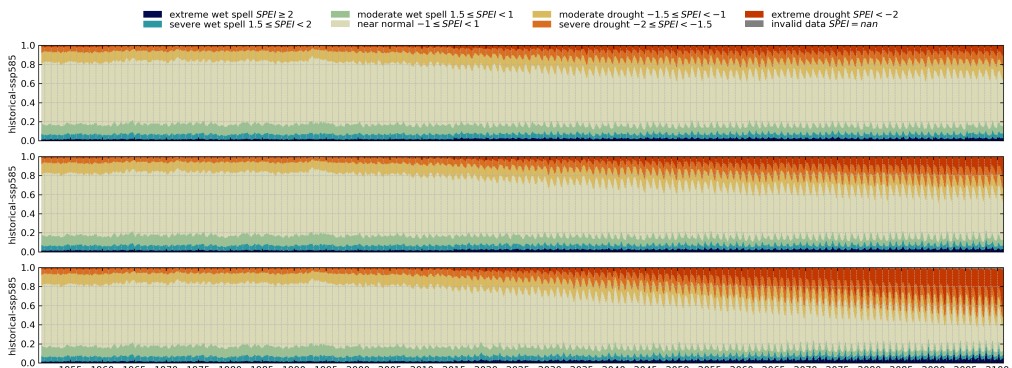

**Figure A7.** Complete time series of relative drought event area in harvest regions according to SPEI. The color indicates how much of the total non glaciated land surface area is covered by certain drought conditions from near normal.

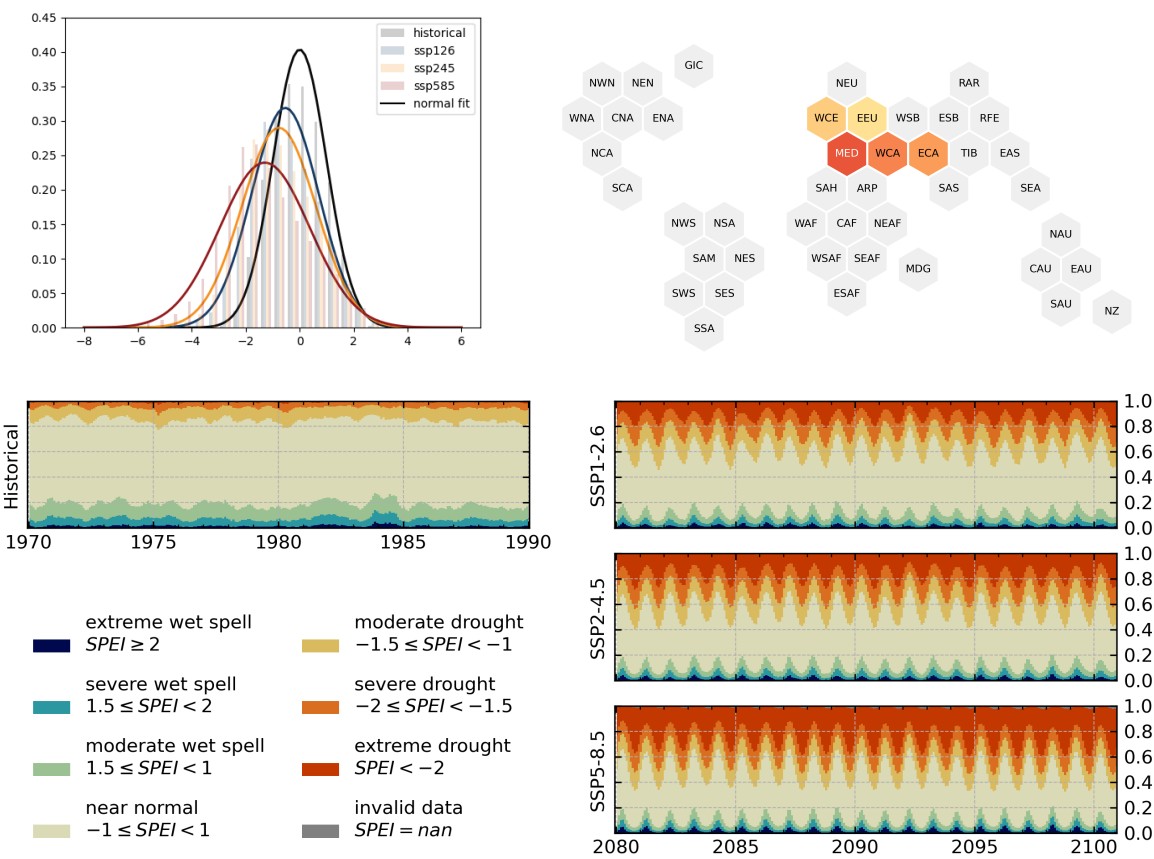

**Figure A8.** Analysis for 5 selected semi arid european and asian regions: WCE, EEU, MED, WCA, ECA.



*Author contributions.* LL conceptualized the study with the help of VE and KW, he conducted the analysis, developed the source code and generated all figures. LL wrote the manuscript with contributions from BH and CJ and feedback from all co-authors.

*Competing interests.* The authors declare that they have no conflict of interest.

*Acknowledgements.* The research for this study was funded by the Deutsche Forschungsgemeinschaft (DFG, German Research Foundation)
through the Gottfried Wilhelm Leibniz Prize awarded to Veronika Eyring (Reference number EY 22/2-1), by the European Research Council (ERC) Synergy Grant "Understanding and modeling the Earth System with Machine Learning (USMILE)" under the EU Horizon 2020 research and innovation program (grant agreement no. 855187) and the European Union's Horizon 2020 research and innovation program under grant agreement 101003536 (ESM2025 – Earth System Models for the Future). LL was supported by the Central Research Development Fund at the University of Bremen with funding no. ZF05/2020/FB1/Causal inference for Earth System Models. We acknowledge
the World Climate Research Programme, which, through its Working Group on Coupled Modeling, coordinated and promoted CMIP, and thank the climate modeling groups for producing and sharing their model outputs, as well as the Earth System Grid Federation (ESGF) for archiving and providing access to the data. This work used resources of the Deutsches Klimarechenzentrum (DKRZ) granted by its Scientific Steering Committee (WLA) under projects no. BD0854 and BD1083.



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
