# Peer review of "Characteristics of Agricultural Droughts in CMIP6 Historical Simulations and Future Projections"

_EGUsphere, 2025_

## Author Comment (AC1)

**Review 1**

This study is highly relevant to the scope of the journal and well-written, making it easy to read. The analysis is comprehensive, and the datasets and methodologies used are appropriate, including innovative approaches. Additionally, I particularly appreciated the high-quality graphics, which enhance the manuscript's clarity and visual appeal. My main concerns involve the lack of detailed information in certain parts of the methodology. Overall, I recommend a minor revision to address these clarifications and provide additional information.

We appreciate the reviewer's thoughtful assessment of our study and we are grateful for the constructive feedback to improve the manuscript. In the revised version of the manuscript, we added information and restructured parts of the methodology. Below, we address each of the reviewer's comments and questions in blue.

Main points:

1. The objectives and added value of the study could have been better framed in the introduction.

Parts of the introduction has been rephrased in the revised version to emphasis our objectives.

2. L154-156: Some additional information on this approximation is needed here.

The actual vapour pressure is approximated using Equation 48 from "Crop evapotranspiration - Guidelines for computing crop water requirements" (Allen et al. 1998). The equation and reference have been added.

3. L179-180: Please provide some information on the methods used by the CRU for the calculation of Eto. If I am not mistaken, they used the Penman-Monteith method. This is just to avoid giving the wrong impression that this is a measured parameter.

Yes, they did, but with slightly different coefficients (short crop assumption). We decided to include another pre-calculated ET0 dataset as some sort of sanity check for our implementation. A sentence has been added for clarity and preventing an impression of ET0 being measured.

4. A sub-section on the methods used to evaluate the CMIP6 models is missing (e.g., pattern correlations, inter-model comparison using root mean squared distance, etc.). Some of this information is provided in the results (e.g., Fig. 1. Caption); however, I recommend providing this information in a brief sub-section of Methods.

We are trying to present all methods, used throughout the study, in a clear and accessible way. Following the reviewers recommendation, a methods subsection for model evaluation has been added, that provides equations and details on our application of pattern correlation, normalized centered root mean squared distance and seasonal cycles.

5. The same applies to the harvest areas. The first information is provided in L255. These need to be specified in the Methods/Data sections, while some context should also be provided in the Introduction.

The paragraphs regarding selection of harvest area (and corresponding figure) have been moved to a dedicated subsection of the methods section. Further, we emphasized our focus on this particular subset of regions in the introduction.

Minor points:

1. L123: The numbering of figures should be consistent with their appearance in the text For example, Fig.6 should be Fig. 1 or just skip the reference to this figure, since you already provide another reference.

Fig 6 became Fig 1, which corrects the order. (See main point 5)

2. L145-150: Please just mention if these projected warming levels are with respect to the preindustrial period.

Yes, they are and since it seems not to be obvious to the potential reader, we added this to the first bullet point.

3. Table 3: Given that the reanalysis/observational datasets are described in the text, I believe this table is redundant and can be removed.

Even though tables can provide better visual guidance, we admit three datasets that have been mentioned before might not be worth their own table. The table has been removed and a reference for the crop mask dataset has been added in the manuscript instead.

4. Figure 5 (6): Please provide the scenarios in the three panels (I assume it is SSP1, SSP2, SSP5 from left to right?) This Figure could also merge with Figure 4 (5).

The reviewer's assumption is correct. There clearly is a lack of information regarding Figure 5 (now 6). The scenario names for each panel have been added to the figure caption. There was meant to be a paragraph between the figures, therefore they should not be merged, unless it improves readability in the final layout.

5. Figure 6: Since these are not results generated in the present analysis, I would move this earlier in the text, where harvest areas and IPCC regions are first mentioned.

The IPCC regions and harvest area selection are moved to a subsection of methods (see main point 5).

6. Figure 7: Please provide units in the y-axis of panel (c). The same applies to panel (b), which I assume is the SPEI value per decade?

Panel (b) is labelled next to the colorbar. It is indeed SPEI/decade. Figure 7 (now 8) has been recreated with additional axis labels for a) and c).

---

## Author Comment (AC2)

**Review 2**

The topic is highly relevant to understanding the evolution of droughts under climate change and could support adaptation strategies, particularly for regions with significant agricultural importance. The paper is well written, the figures are of high quality, and the results are very relevant to climate change studies. I recommend this paper for publication.

Despite this, I have a few minor comments that I would like to see addressed. Overall, I believe there is a lack of transparency in the description of the methodology used for evaluating the models.

We want to thank the reviewer for the invested time and effort and the valuable suggestions for improving the study. In the revised version of the manuscript a subsection has been added to describe the methodology used for the evaluation. In the following we answer each specific comment in blue.

Specific Comments:

1. I found the evaluation of historical simulations to be quite weak. First, the methodology section lacks a clear explanation of how the evaluation was implemented and an explicit definition of all the metrics used, making it difficult to interpret the results, which are also vague. For example, you introduce the variable P in line 195, but you never explicitly state what it refers to. I assume it is the Pearson correlation computed over all grid points (averaged over time), but this is not obvious. Moreover, if I understand correctly, you are only evaluating the models' performance on the time-averaged spatial pattern. I think an additional evaluation of the seasonal cycle and trends would also be important.

To improve transparency in the methodology we added a subsection for model evaluation, which covers calculation details for the distance metrics, pattern correlation and seasonal cycles. Further, we changed its variable name to $R_{pc}$ to avoid any potential confusion with precipitation $P$.

Regarding the second part of the comment: The reviewer correctly pointed out, that only time-averaged evaluations are shown. We agree on the importance of trends and seasonal cycles in a comprehensive evaluation. On the other hand, we would like to keep this study focused on the analysis of future projections. The introduction has been updated to promise a "selective" rather than a "comprehensive" evaluation.

Nevertheless, seasonal cycles are a very important part for our model evaluation and have been added. The simulated seasonal cycles of simulated precipitation are compared to reanalysis and discussed for each of the selected harvest regions. For temperature however, the agreement seems to be high enough to not require a dedicated discussion and the corresponding plots are moved to the appendix.

We did inspect trend maps and global average trends, which generally agreed well for temperature, but less for precipitation. A proper evaluation of trends however, would require a longer time period and would not be in the scope of this study. My colleague, however, is

preparing a study that covers trends in the evaluation of drought indices for longer historical and pre-industrial timeseries.

2. In Figure 1, in panels a and b, you use different names for the same variables. Please make them consistent.

We renamed the variables in panel b) to match the other figures and equations.

3. Line 204: Please explicitly state how the RMSD was calculated.

Equation has been added in methods (see point 1) and referenced here.

4. You do not state the rules for determining whether a model is kept or disregarded based on its performance in historical simulations.

We selected the models based on data availability and limited to one model per institute to not artificially overweight similar models. We did not exclude any of the models based on their performance. Parts of the analysis have been repeated for different subsets of models without much changes to the overall results. However, selecting or weighting the models based on there similarity or performance is an interesting approach, for further studies.

5. Lines 291-292: Is the fraction area calculated with respect to the entire non-glaciated region, following Equation 4? Please be more explicit.

Yes, it is. We added a reference to the equation.

6. Line 294: Please correct the typo: *spacial* into *spatial*

Typo has been corrected.

7. Lines 295-296: This sentence is unclear to me. Please clarify what you mean here.

The sentence has been split and rephrased. Hopefully this makes it more understandable:

"In contrast to spatial mean time series the area fraction plots show the increase in wet and dry conditions at the same time without compensating positive and negative index values. Furthermore, area fraction plots are not influenced by localized extreme SPEI values, as they focus on the proportion of cells exceeding a certain threshold rather than the magnitude of the values themselves, unlike mean time series which can be skewed by a few cells with exceptionally high trends."

8. Lines 303-304: Is this a model-related effect, or do you have another explanation for this? For example, could it be due to increased variability leading to more extreme conditions (both wet and dry)?

Increased variability is very likely to lead to more wet and dry extreme conditions. We also found an increase of extremes (SPEI) during the historical period in observations/reanalysis and models. However, in my opinion the data shown in this study is not sufficient to make such conclusion. My colleague has a study in preparation focusing on the evaluation of SPI and SPEI in the extended historical period, which is going to be submitted soon (also mentioned for point 1). The peaks in summer can be linked to stronger increases of maximum temperature during northern hemisphere summer compared to winter season combined with increasing winter

precipitation in many regions. The shift towards is most likely a side effect of the 6-month accumulation of Index.

9. Line 360: You state that you find similar global trends, but I don't understand where this comment comes from. It seems to me that you are only evaluating the time-averaged spatial patterns. I suspect this is another consequence of the lack of transparency in the methodology and results.

The statement regarding global trend has been removed. The time series in Figure 4 (now 6) included the reanalysis and their trends in an earlier stage of the study. For technical reasons the evaluation part has been separated from the analysis of the observation.

10. Line 370: Please correct *in this scenarios* with *in these scenarios*

Typo has been corrected.